# Broadband albedo of Arctic sea ice from MERIS optical data

Christine Pohl[1], Larysa Istomina[1], Steffen Tietsche[2], Evelyn Jäkel[3], Johannes Stapf[3], Gunnar Spreen[1], and Georg Heygster[1]

[1]Institute of Environmental Physics, University of Bremen, Otto-Hahn-Allee 1, 28359 Bremen, Germany
[2]European Centre for Medium-Range Weather Forecasts, Shinfield Park, Reading, RG2 9AX, United Kingdom
[3]Leipzig Institute for Meteorology, University of Leipzig, Stephanstr. 3, 04103 Leipzig, Germany

**Correspondence:** Christine Pohl (cpohl@iup.physik.uni-bremen.de)

**Abstract.**

Arctic summer sea ice experiences rapid changes in its sea-ice concentration, the surface albedo, and the melt pond fraction. They affect the energy balance of the region and demands an accurate knowledge of those surface characteristics in climate models. In this paper, the broadband albedo (300 – 3000 nm) of Arctic sea ice is derived from Medium Resolution Imaging Spectrometer (MERIS) optical swath data by transforming the spectral albedo as an output from the Melt Pond Detector (MPD) algorithm by a newly developed spectral-to-broadband conversion (STBC). The new STBC replaces the previously applied spectral averaging method to provide a more accurate broadband albedo product which approaches the accuracy of 0.02 – 0.05 required in climate simulations and allows a direct comparison to broadband albedo values from climate models. The STBC is derived empirically from spectral and broadband albedo measurements over landfast ice. It is validated on a variety of simultaneous spectral and broadband field measurements over Arctic sea ice, is compared to existing conversion techniques and performs better than the currently published algorithms. The root mean square deviation (RMSD) between broadband albedo measured and converted by the STBC is 0.02. Other conversion techniques, the spectral averaging method and the linear combination of albedo values from four MERIS channels, achieve higher RMSDs of 0.09 and 0.05, respectively. The improved MERIS derived broadband albedo values are validated with airborne measurements. Results show a smaller RMSD of 0.04 for landfast ice than the RMSD of 0.07 for drifting ice. The MERIS derived broadband albedo is compared to broadband albedo from ERA5 reanalysis to examine the albedo parameterization used in ERA5. Both albedo products agree in the large-scale and temporal pattern. However, consistency in point-to-point comparison is rather poor, with differences up to 0.2, correlations between 0.69 and 0.79, and RMSDs in excess of 0.10. Differences in sea-ice concentration and cloud-masking uncertainties play a role, but most discrepancies can be attributed to climatological sea-ice albedo values used in ERA5. They are not adequate and need revising, in order to better simulate surface heat fluxes in the Arctic. The advantage of the resulting broadband albedo data set from MERIS over other published data sets is the accompanied data set of available melt pond fraction. Melt ponds are the main reason for the sea-ice albedo change in summer but are currently not represented in climate models and atmospheric reanalysis. Additional information on melt evolution together with the accurate albedo can aid the challenging representation of sea-ice optical properties in those models in summer.

# 25 1 Introduction

Broadband albedo of sea ice including the snow covered sea ice is a key parameter in Arctic climate studies. It is defined as the ratio of the reflected to incident irradiance in the waveband 200 - 5000 nm (Pedersen and Winther, 2005; Wendisch and Yang, 2012). The broadband albedo of sea ice determines the energy budget of the surface and controls the heat and mass balance of sea ice cover (Perovich, 1994). Even small changes in the broadband albedo at surface can strongly influence the

Arctic climate by the albedo-feedback mechanism. As broadband albedo of sea ice decreases, more radiation is absorbed at the surface, resulting in an increase of surface temperature and a loss of sea ice which amplifies the albedo reduction (Curry et al., 1995; Perovich et al., 2002; Pirazzini, 2008).

 For a better understanding of the strong sensitivity of the Arctic climate to surface broadband albedo variations, climate studies need surface broadband albedo values which meet two criteria. Firstly, the surface broadband albedo has to be compre-

hensive in space and time to capture its strong variability with low albedo over melt ponds and high albedo over snow covered sea ice. Many surface broadband albedo data products from different satellite missions are available for this purpose. Table 1 provides an overview of the major ones and their specifications. The albedo data from MODerate resolution Imaging Spectroradiometer (MODIS) onboard Terra and Aqua (Lucht et al., 2000; Schaaf et al., 2002) has been widely used as a reference for evaluating other albedo products. The Polar Pathfinder-Extended (APP-x) climate data record from Advanced Very High

Resolution Radiometer (AVHRR) measurements onboard various NOAA POES (U. S. National Oceanic and Atmospheric Administration Polar Operational Environmental Satellites) includes the first long-time series of consistent surface broadband albedo, starting from 1982 (Key et al., 2016). MERIS (Medium Resolution Imaging Spectrometer) onboard ENVISAT (ENVironmental SATellite) provides the surface albedo of Arctic sea ice on a daily basis (Istomina et al., 2015). VIIRS (Visible Infrared Imaging Radiometer Suite) onboard S-NPP (Suomi National Polar-orbiting Partnership) belongs to the newest mod-

erate spatial resolution imaging radiometers generating the surface albedo (He et al., 2018b). High spatial resolution surface albedo (30 m) at the expense of reduced temporal resolution (16 days) are generated by Operational Land Imager (OLI), Thematic Mapper (TM) and Enhanced Thematic Mapper Plus (ETM+) onboard Landsat satellites (He et al., 2018b). A 34 year time series of black-sky surface albedo (SAL) as part of the second edition of the cloud, albedo, and surface radiation data set (CLARA-A2) has been derived from AVHRR measurements onboard NOAA and Metop (Meteorological Operational) satel-

lites (Riihelä et al., 2013; Karlsson et al., 2017). Furthermore, surface albedo can be retrieved from collaborative observations of multiple platforms, e. g., GlobAlbedo (Muller et al., 2012; Lewis et al., 2013) and Global LAnd Surface Satellite (GLASS - Liu et al., 2013; Liang et al., 2013), or from radiation budget data sets, e. g., Clouds and the Earth's Radiant Energy System – Energy Balanced And Filled (CERES-EBAF, 2014; Loeb et al., 2018) and International Satellite Cloud Climatology Project (ISCCP – Zhang et al., 1995, 2004).

Secondly, the surface broadband albedo included in climate models has to fulfill an absolute accuracy of 0.02 – 0.05 as shown in sensitivity analysis (Henderson-Sellers and Wilson, 1983; Jacob and Olioso, 2005; Sellers et al., 1995). However, the satellite retrieved albedo products underlie uncertainties depending on the number and spectral distribution of satellite channels (i. e., narrowbands) available, on the temporal and spatial resolution of the measurement (Tab. 1), as well as on the

approach to retrieve the broadband albedo (kernel driven BRDF (bidirectional reflectance distribution function) based methods or ARC (anisotropy reflectance correction) methods, both combined with a narrow-to-broadband conversion (NTBC), direct estimation methods, and methods using RTT (radiative transfer theory)) (Liang, 2001; Qu et al., 2014). Latter dependency is caused by pre-assumed surface and atmospheric conditions in most retrieval methods. Their representativeness to the actual ones determine the accuracy of the retrieved albedo product. Only the algorithm for MERIS data by Istomina et al. (2015) utilizes a physical model of sea ice in consideration of its melting stage and, therefore, does not depend on a priori assumptions of the surface and atmospheric conditions which can introduce an error in the retrieved albedo product.

However, the retrieval algorithm by Istomina et al. (2015) calculates the broadband albedo from MERIS data by linearly averaging the MERIS derived spectral albedo at wavelengths $400, 500, 600, 700, 800,$ and 900 nm. Therefore, the MERIS derived broadband albedo only covers the waveband 400 - 900 nm. It cannot be used directly in climate studies which usually need surface albedo values in the waveband 300 – 5000 nm (He et al., 2018b) and is not comparable with other satellite derived broadband albedo products with wavebands larger than 300 - 3000 nm (Tab. 1). A comparison of surface broadband albedo products in different wavebands reveals significant discrepancies which can exceed the required albedo accuracy in climate simulations (Li and Leighton, 1992; Winther et al., 2003; Bourgeois et al., 2006). The validation of MERIS derived broadband albedo against airborne measurements showed that over landfast ice the MERIS broadband albedo is higher by 0.06 on average than respective airborne measurements in the waveband 285 – 2800 nm because of its lack of radiative information in the near infrared spectral region (Istomina et al., 2015).

In this paper, we introduce a spectral-to-broadband conversion (STBC) which replaces the conversion method proposed by Istomina et al. (2015) to calculate broadband albedo in the waveband 300 – 3000 nm from spectral albedo retrieved from MERIS swath observations by the Melt Pond Detector (MPD) algorithm (Zege et al., 2015). With these new broadband albedo values, we improve the validation results of the MPD algorithm by Istomina et al. (2015). The new MERIS broadband albedo can be used to evaluate and validate other albedo products from different satellite observations and to test albedo parameterizations in climate models and atmospheric reanalyses. As the MPD algorithm additionally retrieves the melt pond fraction from MERIS observations, it is possible for the first time to include simultaneous derived broadband albedo values and melt pond fractions into climate models and atmospheric reanalyses.

After a short description of the MPD algorithm, the empirically derived spectral-to-broadband conversion of the albedo is presented (Sect. 2). It is tested against the conversion techniques by Istomina et al. (2015) and Gao et al. (2004) in Sect. 3. Section 4 discusses its uncertainty. The new broadband albedo of Arctic sea ice retrieved from MERIS observations is validated in Sect. 5. In Sect. 6, it is compared with respective values derived from the fifth generation of climate atmospheric reanalysis data sets ERA5 produced by the European Centre for Medium-Range Weather Forecasts (ECMWF) (Hersbach et al., 2018) to investigate the consistency of the sea-ice albedo used in a state-of-the-art atmospheric reanalysis with satellite observations. Section 7 concludes our findings.

For reasons of short expressions, the following sections always refer to albedo products at surface even if they are not explicitly declared as such.

## 2 Method

Daily averages of broadband albedo over Arctic sea ice are derived from MERIS observations in three steps: MERIS swath Level 1b data over Arctic sea ice are used by the Melt Pond Detector (MPD) algorithm to derive time-synchronous spectral albedo values which are daily averaged afterward (Sect. 2.1). The output is converted into broadband albedo values (Sect. 2.2).

### 2.1 Melt Pond Detector algorithm

The Melt Pond Detector (MPD) algorithm has been developed by Zege et al. (2015) to retrieve the spectral albedo at wave-
lengths $\lambda_i = 400, 500, 600, 700, 800,$ and 900 nm and the melt pond fraction over Arctic sea ice. It has been validated by Istomina et al. (2015) and the built-in cloud filtering has been improved by Marks (2015). The main retrieval steps of the MPD algorithm are given in this section.

The input parameters are Level 1B swath data with a spatial resolution of 1 km at nadir from MERIS onboard ENVISAT, including radiances of channels 1, 2, 3, 4, 8, 10, 12, 13, and 14 at center wavelengths 412.5, 442.5, 490, 510, 681.25, 753.75,
778.75, 865, and 885 nm, respectively, and solar, and observation angles. Additional relevant information about the atmospheric and surface state (atmosphere profile, aerosol load, bounds for ice and pond optical properties) can be entered from a separate input file.

For each classified sea-ice grid-cell, ice and pond parameters (optical thickness of pond and ice, scattering coefficient of ice, effective ice grain size, and absorption coefficient of yellow pigments) as well as a pond fraction are initialized. From
those, the white ice and melt pond BRDF are calculated based on the asymptotic solution for optically thick layers (Zege et al., 1991). In case of the melt pond BRDF calculation, the optically thick layer is referred to the melt pond bottom. The reflection and transmission at air-water interface is determined by Fresnel's law. The surface BRDF is calculated as a linear combination of both BRDF values weighted by the pond fraction. From surface BRDF and the atmospheric reflectance and transmittance calculated by the radiative transfer code RAY (Tynes et al., 2001), the radiances at MERIS channels at top of atmosphere
are derived based on the atmospheric correction method by Tanré et al. (1983). In an iterative process based on the Newton-Raphson method (Press and Flannery, 1993), the difference of measured and calculated MERIS radiances is minimized as a function of the ice and pond parameters, and of the melt pond fraction.

From resulting ice and pond parameters, the spectral black-sky albedo (directional-hemispherical albedo (Schaepman-Strub et al., 2006)) at wavelengths $\lambda_i = 400, 500, 600, 700, 800,$ and 900 nm are calculated. The output of the MPD algorithm are
the spectral albedo at mentioned six wavelengths, the melt pond fraction and the estimated retrieval error for each MERIS swath data grid-cell. Daily averages are created by gridding and averaging the output of each MERIS swath on a 12.5 km NSIDC (National Snow and Ice Data Center) polar stereographic projection. More details can be found at https://www.seaice.uni-bremen.de/.

## 2.2 Broadband albedo conversion

To convert the six spectral albedo values $\alpha(\lambda)$ at wavelengths $\lambda_i = 400, 500, 600, 700, 800$, and 900 nm to the MERIS broadband albedo, we introduce the spectral-to-broadband conversion (STBC):

$$\alpha_{bb} = k_0 + \sum_{i=1}^{6} k_i \cdot \alpha(\lambda_i). \tag{1}$$

The coefficients $k_i$ are empirically derived from spectral and broadband albedo values measured by Polashenski (2011) on landfast ice close to Barrow, AK, USA, between April and June of the years 2008 and 2009. The objective of the ground-based measurements was to investigate albedo and melt pond evolution during the melt season. The data are available at http://chrispolashenski.com/data.php.

Spectral and broadband albedo were measured along three transect lines of 200 m length each, orientated west-east and located on first-year sea ice 1 km offshore of Barrow, AK, USA (71.366 ° N, 156.542 ° W). The spectral albedo between 350 and 2500 nm was derived from upward and downward irradiance measurements using an Analytical Spectral Devices (ASD) spectrometer 'FieldSpec3' with a diffuse cosine collector. The broadband albedo in the waveband $300 - 3000$ nm was measured by a Kipp&Zonen CM14 albedometer. Each of the two optics was mounted on a 1.5 m long arm and was positioned about 1 m above the surface (Polashenski, 2011).

Spectral and broadband albedo were measured along a first transect line on 21 days between April and June 2008. Along the second transect line, measurements were done on 13 days between May and June 2009 and along the third transect line on 12 days in June 2009. The second and third transect lines were 1 km apart of each other. The measurements were done once per day (except 14 June 2008 with twice per day) in steps of $2.5 - 5$ m under various sky conditions (clear, cloudy, and overcast skies). The time of spectral and broadband albedo measurements may differ by more than one hour because the spectral and broadband albedo samples were recorded one after the other.

After excluding unphysically high and discontinuous albedo spectra, 1720 albedo spectra and broadband albedo values of various sea-ice surface types are available, from optically thick (snow depth $\gtrsim 10\,\mathrm{cm}$) and thin snow surfaces (snow depth $\lesssim 10\,\mathrm{cm}$), bare ice with loose decaying ice crystals on top, slushy bare ice to frozen and liquid melt ponds. Spatially and temporally collocated spectral and broadband albedo measurements were used to derive the coefficients $k_i$ in Eq. (1) minimizing the square sum of errors $\left( \alpha_{bb} - k_0 - \sum_{i=1}^{6} k_i \cdot \alpha(\lambda_i) \right)^2$. The coefficient $k_0$ was preset to 0 in order to warrant the condition $\alpha_{bb} = \alpha(\lambda_i) = 0$. The resulting coefficients are given in Tab. 2 and can be used to calculate the broadband albedo in waveband $300 - 3000$ nm. This method is referred to as empirically derived STBC.

The daily averages of MERIS derived broadband albedo over Arctic sea ice are available at https://seaice.uni-bremen.de/data/meris/gridded_cldscr/broadband_albedo/.

## 3 Results

We investigate the performance of the empirically derived STBC (Sect. 3.1) and compare this new conversion to those methods
proposed in Istomina et al. (2015) and Gao et al. (2004) (Sect. 3.2). For the evaluation, we used the root mean square deviation
(RMSD) as an estimator for the statistic deviation, the mean difference of retrieved and measured albedo (bias) as an estimator
for the systematic deviation, and the coefficient of determination ($R^2$) which allows conclusions about the quality of correlation.
Measurement uncertainties in the considered data sets are neglected as they are small compared to influences due to time offsets
or different spatial resolutions of the measurements.

### 3.1 Regression analysis of the empirically derived STBC

The performance of the empirically derived STBC by using the training data set is presented in Fig. 1. It shows the broadband
albedo measured by Polashenski (2011) over landfast ice close to Barrow, AK, USA between April and June of the years 2008
and 2009 and the broadband albedo product $\alpha_{\mathrm{bb}}$ retrieved from the empirically derived STBC using respective spectral albedo
values measured by Polashenski (2011). The combination of a high determination coefficient ($R^2 = 0.96$) and a low RMSD of
0.05 between the measured and the retrieved broadband albedo indicates a good performance of the empirically derived STBC.
Systematic errors are not found (bias $\approx 0.00$). The slope of 0.93 and the intercept of 0.04 of the regression line indicate a slight
overestimation of low albedo values and a slight underestimation of high albedo values by the empirically derived STBC.

Note that the deviations between measured and retrieved broadband albedo are not only caused by the empirically derived
STBC but also by time offsets between the measured spectral and broadband albedo. Within this time difference, the pre-
vailing illumination can be significantly modified due to changes in the solar zenith angle and cloud conditions. As a result,
nonconforming spectral and broadband albedo values were measured.

### 3.2 Comparison to existing conversion methods

The empirically derived STBC is compared with conversion methods proposed in Istomina et al. (2015) and Gao et al. (2004).
Their coefficients $k_i$ are listed in Tab. 2. Istomina et al. (2015) calculated broadband albedo by averaging the spectral albedo
values. Gao et al. (2004) propose a linear combination method of albedo values from MERIS channels 3, 5, 7, and 13 at the
center wavelengths $\lambda_i^* = 490, 560, 665$, and 865 nm instead of $\lambda_i$. The coefficients $k_i$ in Gao et al. (2004) were adopted from the
narrow-to-broadband conversion developed for albedo values from MISR (multi-angle imaging spectrometer) onboard Terra
satellite by Liang et al. (1999). This conversion is based on radiative transfer simulations.

For the comparison, an independent test data set of measured spectral and broadband albedo was obtained during the
ACLOUD (Arctic CLoud Observations Using airborne measurements during polar Day) campaign (Wendisch et al., 2019).
The measured spectral albedo is available in Jäkel et al. (2018), details on measured broadband albedo are given in Jäkel
et al. (2019). The meteorological conditions during the campaign are documented in Knudsen et al. (2018). On 25 June 2017,
spectral and broadband albedo were measured on board the aircraft Polar 5 north of Svalbard (80.48 ° – 81.62 ° N, 11.31 ° –

20.36 ° E) over sea ice floes of different sizes under cloud free conditions. The flight altitude during the selected flight sections was between 50 and 200 m.

Spectral albedo was derived from hemispheric up- and downwelling spectral radiation detected by SMART (Spectral Modular Airborne Radiation Measurement System) in the wavelength range 350 – 2200 nm within an accuracy of 10 %. SMART was actively stabilized in a horizontal position (Wendisch et al., 2001; Ehrlich et al., 2008; Bierwirth et al., 2009). The temporal resolution of the spectral measurements was 2 Hz. Broadband albedo in waveband 200 – 3600 nm was measured by two Kipp&Zonen CMP-22 pyranometers within an error of 3 %, as given by the manufacturer for stationary operation. Both pyranometers were mounted in a fixed position at the top and at the bottom of the aircraft's fuselage for a simultaneous detection of the hemispheric down- and upwelling radiation. The temporal sampling interval was increased to 20 Hz by the deconvolution method described in Ehrlich and Wendisch (2015). For clear-sky conditions, the measurements of the upward-facing pyranometer were corrected for deviations from horizontal attitude due to misalignment of the instrument and roll and pitch angles of the aircraft based on a method given by Bannehr and Schwiesow (1993) and described in Lampert et al. (2012). Broadband measurements at roll and pitch angles higher than 4 ° are excluded.

Spectral and broadband albedo values were temporally adjusted. Influences caused by their different sampling intervals (2 Hz and 20 Hz) were diminished by convolving the broadband albedo values by a running average of 10 measurements.

In Fig. 2(a), measured broadband albedo is compared to the broadband albedo $\alpha_{bb}$ retrieved by the empirically derived STBC using spectral albedo measured by SMART. Retrieved and measured broadband albedo are highly correlated ($R^2 = 0.98$). The regression slope of 1.06 is close to 1 and together with the -0.03 intercept causes a small underestimation of low albedo values. Discrepancies between both broadband albedo products ($\mathrm{RMSD} = 0.02$) are caused by uncertainties in the empirically derived STBC (Fig. 1) as well as by measurement uncertainties in the spectral and broadband albedo.

For comparison, Figs. 2(b) and (c) show measured broadband albedo and respective retrieved broadband albedo $\alpha_{bb}$ with conversion methods proposed in Istomina et al. (2015) and Gao et al. (2004), respectively. Retrieved broadband albedo values of both conversions correlate well with measured broadband albedo values ($R^2 = 0.98$ in both cases). But the conversion from Istomina et al. (2015) overestimates the broadband albedo (mean difference of $\mathrm{bias} = 0.08$) while the conversion from Gao et al. (2004) underestimates it ($\mathrm{bias} = -0.04$). Accordingly, RMSD values (0.09 and 0.05) are higher than for the empirically derived STBC.

## 4 Uncertainties due to differing variables along the spectral-to-broadband conversion

Along the path of converting the spectral albedo to broadband albedo, inconsistencies of physical (state of the sea-ice surface, solar zenith angle, waveband, contrast between black- and blue-sky albedo) and meteorological variables (sky and atmospheric condition) may influence the accuracy of the derived broadband albedo product. Their presence and impact on the albedo is discussed in the following.

The STBC is based on various sea-ice surface types in the early season (April - June) mentioned at the end of Sect. 2.2. Nevertheless, it can be applied to later seasonal (July-September) Arctic surface type as well, as those surface types like snow

covered sea-ice, white ice, and refrozen surface types are similar to the surface types included in the training data-set (snow layers and frozen melt ponds). Therefore, the STBC can be used for all Arctic sea-ice surface types.

Moreover, we have tested the performance of a STBC, when it is derived for each surface type separately. For that, we classified the albedo measurements by Polashenski (2011) in those over "snow surface", "bare ice", and "melt pond". Measurements from 2009 (1194 samples) were used as a training data set to derive the coefficients $k_i$ in Eq. 1 for each surface class. The surface specific STBCs were tested with the respective classified albedo measurements from 2008 (526 samples). The RMSD, bias, and $R^2$ between measured and retrieved broadband albedo are 0.05, 0.02, 0.96, respectively. Using the albedo measurements from 2009 independent of the surface type as the training data set to derive the coefficients $k_i$ in Eq. 1 and applying those to all albedo data from 2008 leads to the same RMSD, bias and $R^2$. Therefore, the STBC can be derived independently of the Arctic surface type without deterioration of their accuracy.

A substantial uncertainty in the derivation of the broadband albedo is the sky condition. Most of the albedo measurements used for the empirical development of the STBC were influenced by clouds. Those can slightly increase the broadband albedo of snow by up to 0.06 (Key et al., 2001; Grenfell and Perovich, 1984) and, therefore, influences the empirical derivation of the STBC. This cloud influenced STBC is applied in Sect. 3.2 to albedo measurements under clear-sky conditions which can contribute to the discrepancies shown in Fig. 2(a). However, deriving the coefficients in Eq. (1) using only albedo measurements under clear-sky conditions (113 albedo spectra and broadband albedo values) by Polashenski (2011) and an application of those to the cloud free albedo measurements during ACLOUD enlarges the RMSD and bias given in Fig. 2(a) by 0.02 and 0.03, respectively (not shown).

According to Liang (2001), a wide range of atmospheric conditions and solar zenith angles are required to develop the STBC generally applicable to any spectral albedo measurement in the Arctic. We can confirm a high variability of the former via available aerosol optical depth from AERONET (Aerosol Robotic Network) data measured in Barrow, AK, USA, for the days of albedo measurements by Polashenski (2011) when the sun disk was visible. However, due to the lack of the exact times of albedo measurements, we cannot determine the actual variability in the solar zenith angle. Since variations in the atmospheric visibility and the solar zenith angle influence the broadband albedo (Liang, 2001) only marginally, they play a minor role in the uncertainties of the empirically derived STBC. Additionally, those effects are overlaid by the cloud effects mentioned above.

Different wavebands of broadband albedo products may affect their comparison. Applying the STBC, the derived broadband albedo covers the wavelength range 300 – 3000 nm. In contrast, the broadband albedo at ACLOUD was measured in a broader waveband (200 – 3600 nm). Resulting discrepancies between both albedo products are estimated by the radiative transfer model libRadtran (Mayer and Kylling, 2005) and are lower than 0.01. Those systematic errors are negligible when comparing both broadband albedo products.

Although the STBC has been derived for blue-sky albedo (bihemispherical reflectance (Schaepman-Strub et al., 2006)) in Sect. 2.2, it shall be applied to the MERIS derived spectral albedo which is referred to as black-sky albedo (directional-hemispherical reflectance). Manninen et al. (2012) estimated the discrepancy between black- and blue-sky broadband albedo for 87 individual surface reflectance spectra from USGS (U. S. Geological Survey) Spectroscopy Lab data which includes, i. a.,

snow, ice and water spectra. It is typically lower than 0.05 at solar zenith angles around 60 ° and at aerosol optical depths up to 1 at 440 nm. We expect that the aerosol optical depth is considerably lower in the Arctic such that the difference between black- and blue-sky albedo decreases and seems to be negligible against the cloud effect. Hence, we decide not to adjust any albedo measurement for the blue-sky / black-sky discrepancy.

## 5 Improved Melt Pond Detector Validation

The MPD algorithm by Zege et al. (2015) is combined with the empirically derived STBC to retrieve more accurate broadband albedo values of Arctic sea ice from MERIS observations compared to the procedure described in Istomina et al. (2015). With these new values, we will improve the validation of the MPD algorithm performed in Istomina et al. (2015).

The new MERIS broadband albedo data product is validated with measured broadband albedo during the aircraft campaign MELTEX (Impact of melt ponds on energy and momentum fluxes between atmosphere and sea ice) performed by the Alfred Wegener Institute, Helmholtz Centre for Polar and Marine Research (AWI) over the southern Beaufort Sea between May and June 2008. The goal of the campaign was to improve the quantitative understanding of the impact of melt ponds on radiation, heat and momentum fluxes over Arctic sea ice (Birnbaum et al., 2009).

To determine the broadband albedo, two Eppley Precision Spectral Pyranometers were mounted on aircraft Polar 5 in a fixed position measuring simultaneously the hemispheric down- and upwelling radiation integrated over the waveband from 285 to 2800 nm. Analogous to the broadband measurements during ACLOUD, the measurements of downwelling radiation at clear-sky conditions were corrected for deviations from horizontal attitude based on the method given by Bannehr and Schwiesow (1993).

On five days (26 May, 3, 4, 6, and 7 June 2008), airborne broadband albedo was measured under almost clear-sky conditions. Measurement times are given in Tab. 3. Maps of flight tracks are shown in Birnbaum et al. (2009). The flight altitude was between 50 and 400 m. On 6 June, broadband albedo was measured over landfast ice, on the other selected days measurements were performed over drifting ice with ice floes of different sizes.

Due to two warming events between 23 – 24 May and 1 – 7 June, broadband albedo was measured over sea ice at different melting stages: On 26 May and 3 June, melt ponds at early stages were overflown, on 4 June the snow was either melted away or was very wet and the number and size of melt ponds rose. On 6 and 7 June, the observed sea ice was homogeneously covered by well developed melt ponds.

Airborne measured broadband albedo is collocated with MERIS broadband albedo derived for each grid-cell from MERIS swath data at reduced resolution of $1.2 \, \mathrm{km}$. The MERIS overflight times which corresponds to the airborne flights are also given in Tab. 3. The time difference between the airborne and satellite measurements stays below 2 hours except for 3 and 7 June with a maximum difference of 3 and 4 hours, respectively.

For the collocation of airborne and MERIS broadband albedo, the orthodromic distance between each MERIS grid-cell center and the location of a given airborne measurement is calculated. Each airborne measurement is allocated to that satellite grid-cell with the least orthodromic distance. Depending on orientation and location of flight track as well as MERIS grid-cell

availability, up to 14 airborne broadband albedo measurements are collocated to one MERIS derived broadband albedo value. The airborne broadband albedo values are averaged for each satellite grid-cell.

The scatterplots in Fig. 3 show the correlation of the MERIS derived and collocated airborne broadband albedo for measurements over landfast ice (a) and over drift ice (c), respectively. Collocated airborne and MERIS broadband albedo values are used for validation only if more than five airborne broadband albedo values are available for one MERIS grid-cell. Time

differences between airborne and MERIS measurements up to 1.5 h were allowed.

For the landfast ice case (Fig. 3(a)), MERIS broadband albedo is slightly higher at low albedo values than the corresponding airborne broadband albedo. For drift ice cases (Fig. 3(c)), the respective overestimation is higher but at high albedo values an underestimation occurs (slope $= 0.53$, intercept $= 0.25$).

The highest correlation and lowest RMSD value is found for broadband albedo values over landfast ice on 6 June 2008

($R^2 = 0.93$, RMSD $= 0.04$, Fig. 3(a)). Compared to landfast ice, the RMSD $= 0.07$ for the drifting ice cases (Fig. 3(c)) is nearly twice as high and the correlation is lower ($R^2 = 0.78$). Correlation and RMSD values depend on weather conditions, ice concentration, surface inhomogeneity, time differences between satellite overflight and airborne measurements, and spatial resolution of the observation. As landfast ice is stationary over long time, airborne measurements are not displaced relative to satellite measurements during the time lag between both observations. Over drifting ice floes at lower sea-ice concentrations, a

higher sea-ice drift can be expected (free drift conditions). Therefore, the surface conditions observed by aircraft and satellite can get more different (see the three outliers at low measured albedo values in Fig. 3(c)). The discrepancies between MERIS derived and airborne broadband albedo values are increased by their different spatial resolution which depends on observation heights and instrument optics. As clearly seen in the histogram in Fig. 3(d) the coarser spatial resolution of MERIS observations leads to a lower albedo variability compared to the respective airborne measurements. Moreover, the bimodal albedo distribu-

tion shown in the latter are smeared out in the respective satellite retrievals. For the landfast ice case, the surface conditions are more homogeneous and thus the difference in spatial resolution has less influence on the comparison (see Fig. 3(b)).

In contrast, the validation of MERIS broadband albedo values derived by the procedure described in Istomina et al. (2015), i. e., the combination of the MPD algorithm and the spectral averaging method by coefficients given in Tab. 2, results in similar correlation values of $R^2 = 0.94$ for the landfast ice case and $R^2 = 0.78$ for the drift ice cases. The RMSDs are 0.06 (landfast

ice) and 0.09 (drift ice) and thus exceed the values found above. This again illustrates the better performance of the empirically developed STBC compared to the spectral averaging method proposed in Istomina et al. (2015).

## 6   Comparison between broadband albedo from satellite and atmospheric reanalysis

During the melt season in Arctic summer, the Arctic sea ice experiences rapid changes in the albedo mainly caused by the melt pond development (Eicken et al., 2002; Polashenski, 2011). As the melt pond fraction on sea ice evolves according to

the sequence of melt stages (Eicken et al., 2002), the melt pond fraction and the melt evolution are beneficial information in the representation of the summer energy balance in atmospheric reanalyses, especially in above mentioned situations where accurate albedo representation is challenging.

Although a vast amount of satellite and in-situ observations are assimilated into every atmospheric reanalysis, there is no published reference about assimilation of melt ponds in those models at the time of writing. Instead, the sea-ice surface is
characterized by a parameterized surface albedo. We test the albedo parameterization used for the atmospheric reanalysis data set ERA5 by comparing ERA5 and MERIS broadband albedo values.

ERA5 is the fifth generation of climate reanalysis data sets produced by ECMWF. It provides hourly values of atmospheric and land variables from 1950 onwards with a spatial resolution of 31 km, obtained by an ensemble 4D-Var data assimilation system (Hersbach et al., 2018). The sea-ice albedo is simply prescribed as constants following the albedo values given in Ebert
and Curry (1993) and considers the spectral variation. It is representative for dry snow during the months September to May, representative for melting snow in June, and representative for bare sea ice in July and August. Melt ponds are not represented at any time. The Ebert and Curry values are taken to be valid at the 15th day of the month, and the values for all other days are obtained by linear interpolation. The grid-cell albedo for ERA5 is then a linear combination between this sea-ice albedo, and the open water albedo:

$$\alpha_{\text{ERA5}} = c \cdot \alpha_{\text{seaice}} + (1 - c) \cdot \alpha_{\text{water}} \quad , \tag{2}$$

with $c$ representing the sea-ice concentration from ERA5 generated by Copernicus Climate Change Service (2017). It has been developed under the Ocean and Sea Ice Satellite Application Facilities (OSI-SAF) using passive microwave satellite data from SSMIS (Special Sensor Microwave Imager Sounder) at 10 km resolution (Tonboe et al., 2017). The open water albedo $\alpha_{\text{water}}$ is calculated in dependence of the solar zenith angle according to Taylor et al. (1996) for the clear-sky case and is equal
to $\alpha_{\text{water}} = 0.06$ for diffuse irradiation (ECMWF, 2016).

The effective grid-cell broadband albedo of ERA5 can be calculated from hourly averaged ERA5 net and downward irradiances $F_{\text{net}}(h)$ and $F_{\text{dn}}(h)$ at surface in the waveband $200 - 4000$ nm. Thus, daily averaged ERA5 grid-cell broadband albedo is given by:

$$\alpha_{\text{ERA5}} = 1 - \frac{\overline{F_{\text{net}}(h)}}{\overline{F_{\text{dn}}(h)}} \quad . \tag{3}$$

The bars illustrate the averaging of the hourly ($h$) irradiances per day at which the sun is above the horizon. $F_{\text{net}}$ and $F_{\text{dn}}$ were generated by using Copernicus Climate Change Service (2017).

Analogously, the MERIS grid-cell broadband albedo for sea-ice grid-cells can be expressed as:

$$\alpha_{\text{MERIS}} = (1 - s) \cdot \alpha_{\text{ice}} + s \cdot \alpha_{\text{pond}} \quad , \tag{4}$$

since the STBC is linear. The parameters $\alpha_{\text{ice}}$ and $\alpha_{\text{pond}}$ represent the broadband albedo of ice and melt pond, and $s$
represents the melt pond fraction.

Although the structure of Eqs. (2) and (4) are identical, the single variables cannot be compared directly. According to the MPD algorithm, MERIS derived albedo $\alpha_{\text{MERIS}}$ is only available at 100 % sea-ice concentration (Sect. 2.1) and is simultaneous

derived by the melt pond fraction $s$ and albedo portions $\alpha_{\mathrm{ice}}$ and $\alpha_{\mathrm{pond}}$. Consequently, all three components are influenced by the reflection properties of undetected small cracks and leads (oceanic water). In contrast, the ERA5 albedo $\alpha_{\mathrm{ERA5}}$ does not consider melt ponds and their components $c$, $\alpha_{\mathrm{seaice}}$, and $\alpha_{\mathrm{water}}$ are determined independently. Thus, unconsidered melt ponds influence the sea-ice concentration $c$ only and in a non-linear relationship. However, both equations can explain variations and differences between the ERA5 and MERIS grid-cell broadband albedo which are considered below.

## 6.1 Temporal differences over pure sea ice

To understand the sea-ice albedo parameterization in ERA5, we illustrate the seasonal variability of the broadband albedo $\alpha_{\mathrm{ERA5}}$ at an ERA5 sea-ice concentration of $c = 1$ in Fig. 4 (a). Thus, Eq. (2) modifies to $\alpha_{\mathrm{ERA5}} = \alpha_{\mathrm{seaice}}$. Only grid-cells with a minimum distance of 50 km from coastlines are taken into account. The ERA5 values are averaged spatially and yearly over the years 2003 to 2011. The literature based values for dry snow on 15 May and 15 September (0.84 - 0.85), for melting snow on 15 June (0.7) and for bare sea ice between 15 July and 15 August (0.55 - 0.57) can be clearly identified. Intermediate values are linearly interpolated. Small fluctuations of the ERA5 albedo of up to 0.05 around the prescribed constant values are most likely a consequence of diagnosing the broadband albedo from net and downwelling irradiances at the surface (Eq. (3)): varying amounts of water vapour above the ice cause variations in the spectral composition of downwelling irradiance $F_{\mathrm{dn}}$, and the spectral dependence of the ERA5 albedo values then leads to an apparently varying broadband albedo.

The corresponding averaged MERIS broadband albedo $\alpha_{\mathrm{MERIS}}$ and melt pond fraction $s$ are comparably shown in Fig. 4 (a). The averaged MERIS albedo $\alpha_{\mathrm{MERIS}}$ possesses a similar seasonal development compared to the ERA5 albedo $\alpha_{\mathrm{ERA5}}$. However, the MERIS albedo is lower in May to begin of June by up to 0.06 as well as from end of August to September by up to 0.11. Between 10 June and 22 August, the MERIS albedo is up to 0.11 higher compared to the ERA5 albedo. The averaged melt pond fraction from MERIS is between 0.06 (May, September) and 0.21 (July, begin of August). The MERIS products offer a higher variability than the ERA5 broadband albedo, indicated by a higher standard deviation and stronger day to day variations. It is highest in July and August when the Arctic surface exhibit the highest heterogeneity due to melting processes.

In May, the melt pond fraction is overestimated since the temperatures are below the melting point (Fig. 4 (b)). Unscreened clouds which are darker than the surface are treated as melt ponds by the MPD algorithm (Sect. 2.1) leading to melt pond fractions between 0.02 and 0.15 and a reduced MERIS broadband albedo $\alpha_{\mathrm{MERIS}}$. After melt onset in June, the Arctic surface is darker than clouds and therefore distinguishable from those. Thus, the derived melt pond fractions in July and August are reliable. The conflicting ERA5 sea-ice concentration of $c = 1$ might be due to different spatial resolutions in the input data set (1 km for MERIS, 10 km for SSMIS): small total melt pond areas, e. g., 0.1 $\mathrm{km}^2$, equal a melt pond fraction of $s = 0.1$ in a MERIS grid-cell but only $s = 0.001$ in a SSMIS grid-cell and cannot be detected by SSMIS. Additionally, values of $c = 1$ in July are questionable as the 2 m temperature is above the freezing point (Fig. 4 (b)) implicating wet surfaces which are detected immediately by passive microwave instruments. The modified signals result only in a retrieved sea-ice concentration of $c = 1$, if the deviation of the sea-ice emissivity in the observed grid-cell is outside the assumed variability around the tie point in the retrieval algorithm.

Despite existing melt ponds in a grid-cell from end of June to end of August, the MERIS derived albedo $\alpha_{\mathrm{MERIS}}$ can be higher than the broadband albedo from ERA5 $\alpha_{\mathrm{ERA5}}$. However, the latter remains within the standard deviation of $\alpha_{\mathrm{MERIS}}$,

implying that the literature based albedo value of bare ice $\alpha_{\mathrm{seaice}}$ is reasonable but slightly low-biased for surface types consisting of a mixture of melt ponds, wet snow, and bare ice. From mid-September after refreezing onset, the passive microwave instrument (SSMIS) retrieving the ERA5 sea-ice concentration cannot detect any wet surfaces whereas the optical instrument (MERIS) identifies optically darker frozen melt ponds ($s \approx 0.06$). Consequently, the grid-cell broadband albedo from ERA5 is only determined by the dry snow albedo and is therefore overestimated. Due to the temporal linear interpolation, the ERA5

broadband albedo $\alpha_{\mathrm{ERA5}}$ overestimation already starts from end of August.

## 6.2   Temporal differences over first-year and multiyear ice

Figures 5 (a) and (b) show the time series of the broadband albedo of MERIS $\alpha_{\mathrm{MERIS}}$ and ERA5 $\alpha_{\mathrm{ERA5}}$ as well as the melt pond fraction $s$ and the complement of the sea-ice fraction $c$ for a first-year ice area in the Beaufort Sea close to Barrow, AK, USA (75 ° N, 155 ° W) and for a multiyear ice area north of Greenland (84.5 ° N, 35 ° W) in summer 2007, respectively. Each

area is composed of five grid-cells. The seasonal variability of the four products in both areas averaged over the years 2003 to 2011 are shown in Figs. 5 (c) and (d). The MERIS and ERA5 grid-cell broadband albedo agree in seasonal pattern, and the albedo decreases with increasing melt pond fraction and complement of sea-ice fraction and vice versa, especially clearly seen at the area over multiyear ice (Figs. 5 (b) and (d)) and the short melt and refreeze event on first-year ice around 9 June 2007 (Fig. 5 (a)). However, temporal differences between MERIS and ERA5 grid-cell broadband albedo occur, which are highest

over first-year ice starting from mid-August with values up to 0.2 and over multiyear ice between June and August with values around 0.1.

The overestimation of the melt pond fraction in May, likely caused by clouds, is higher over first-year ice than over multiyear ice resulting in a MERIS grid-cell broadband albedo underestimation over first year ice in 2007 (Fig. 5 (a)) and in the nine year average (Fig. 5 (c)) whereas over multiyear ice, both grid-cell broadband albedo products coincide (Figs. 5 (b) and (d)).

Between end of May and mid-June, the ERA5 grid-cell broadband albedo is underestimated over multiyear ice by up to 0.1 (Figs. 5 (b) and (d)) due to the at this time prevailing ERA5 sea-ice albedo $\alpha_{\mathrm{seaice}}$ reduction (Fig. 4 (a)). This reduction is strong enough to obliterate grid-cell albedo affects by small variations in the complement of ERA5 sea-ice concentration at 6 June (Fig. 5 (b)). In contrast, the ERA5 sea-ice albedo $\alpha_{\mathrm{seaice}}$ reduction is unremarkable in the comparison of the ERA5 and MERIS grid-cell broadband albedo over first-year ice because it is overlaid by strong variations in the complement of

ERA5 sea-ice concentration at begin of June as well as the underestimation of the MERIS grid-cell broadband albedo product in May. As a result, both grid-cell broadband albedo products agree over first-year ice in 2007 and in the nine year average at the begin of June. The high correlation between the melt pond fraction from MERIS and the complement of the ERA5 sea-ice concentration, e. g., over first-year and multiyear ice in June 2007 shows the influence of melt ponds on the sea-ice concentration in ERA5.

From mid-June to mid-July, both grid-cell broadband albedo products coincide well over first-year ice in the year 2007 whereat in the nine year average, the increase of the melt pond fraction leads to up to 0.1 smaller MERIS grid-cell broadband

albedo values in June. Starting from July 2007, melting processes lead to strong variations in the MERIS products, especially seen on two peaks at 23 July and 8 August (Fig. 5 (a)). At both times, the ERA5 grid-cell broadband albedo follows the MERIS grid-cell broadband albedo but its variation stays low. When the complement of ERA5 sea-ice concentration reaches 1 starting

from end of August 2007, MERIS products are not available because the MPD algorithm does not detect any sea ice. In the nine year average, ERA5 grid-cell broadband albedo is higher than MERIS grid-cell broadband albedo over first-year ice between mid-July and mid-September. Its minimum albedo is limited to 0.2 due to the literature based ERA5 bare ice albedo $\alpha_{\mathrm{seaice}}$ and prevailing averaged ERA5 sea-ice concentration $c$. MERIS grid-cell broadband albedo can be even lower despite of low melt pond fractions as melting sea ice can be very dark. Starting from mid-September, both averaged grid-cell broadband albedo

products increases due to refreezing processes. As the ERA5 albedo $\alpha_{\mathrm{ERA5}}$ is only tuned by the ERA5 sea-ice concentration (compare with Fig. 4 (a)), its increase is weaker than those from MERIS broadband albedo $\alpha_{\mathrm{MERIS}}$.

In contrast to first-year ice, the grid-cell broadband albedo from MERIS over multiyear ice is mostly higher by up to 0.1 than from ERA5 between mid-June and end of August (Figs. 5 (b) and (d)). Two reasons can be identified for this effect. Firstly, the ERA5 sea ice albedo $\alpha_{\mathrm{seaice}}$ is lower than the MERIS albedo $\alpha_{\mathrm{MERIS}}$ (Fig. 4 (a)). Secondly, water albedo $\alpha_{\mathrm{water}}$ is much

lower than albedo of melt ponds $\alpha_{\mathrm{pond}}$ such that even lower values of the complement of ERA5 sea-ice concentration than of the melt pond fraction cannot compensate the low ERA5 water albedo. Contrary to the small variations in the complement of the ERA5 sea-ice concentration at 6 June, its abrupt changes at 1 July and 7 August 2007 are reproduced in the ERA5 albedo $\alpha_{\mathrm{ERA5}}$ (Fig. 5 (b)) as the variation in the complement of the sea-ice concentration is stronger and, in case of 7 August, the ERA5 sea-ice albedo $\alpha_{\mathrm{seaice}}$ keeps constant (Fig. 4 (a)). However, the strong variability in MERIS broadband albedo $\alpha_{\mathrm{MERIS}}$

at the begin of August 2007 is not reflected in the ERA5 broadband albedo $\alpha_{\mathrm{ERA5}}$.

Towards the begin of September, the ERA5 sea-ice concentration $c$ and sea-ice albedo $\alpha_{\mathrm{seaice}}$ of multiyear ice increase which compensates the difference between ERA5 and MERIS grid-cell broadband albedo. However, in the nine year average, lower MERIS grid-cell broadband albedo are observed as frozen surfaces can be optically darker (Sect. 6.1). In the second half of September, MERIS products are no more available because of clouds and too low solar zenith angles.

## 6.3   Spatial differences during melt onset

Figure 6 compares daily averaged ERA5 and MERIS grid-cell broadband albedo of Arctic sea ice on three days in June 2007 during melt onset. Cloud contaminated grid cells in MERIS data are flagged by MPD algorithm and are removed. On 2 June 2007, just few days before melt onset, ERA5 albedo $\alpha_{\mathrm{ERA5}}$ is mostly smaller than the MERIS albedo $\alpha_{\mathrm{MERIS}}$ by up to 0.1 except in the Chukchi Sea, East Siberian Sea, and southern Beaufort Sea. As discussed in Sect. 6.2, the ERA5 albedo

parameterization with too low sea-ice albedo values $\alpha_{\mathrm{seaice}}$ is responsible for these differences (compare with Fig. 5 (d)). In the Chukchi Sea, East Siberian Sea, and southern Beaufort Sea the melt pond fraction is higher ($0.2 \leq s \leq 0.37$) than in other Arctic regions ($s < 0.13$, not shown), which leads to MERIS albedo $\alpha_{\mathrm{MERIS}}$ smaller by up to 0.27 than respective ERA5 albedo values $\alpha_{\mathrm{ERA5}}$. On 12 June 2007, the melting season has started between 120 ° W and 120 ° E which is clearly visible by lower ERA5 and MERIS grid-cell broadband albedo values (0.30 – 0.60). Although the decrease in the albedo is mainly

regulated by different parameters (Eq. (2) and (4)), the resulting albedo values are similar. North of Greenland, the Fram Strait,

and Svalbard, ERA5 grid-cell broadband albedo remains lower compared to MERIS by values up to 0.19 whereas it is the opposite between 105 ° E and 160 ° E, as well as in the Chukchi Sea, East Siberian Sea, and southern Beaufort Sea. Negative differences are caused by the combination of too low ERA5 albedo values of sea ice $\alpha_{\mathrm{seaice}}$ and water $\alpha_{\mathrm{water}}$ (compare with Fig. 5 (d)). Positive differences are caused by the melt ponds which reduce the MERIS broadband albedo $\alpha_{\mathrm{MERIS}}$ (compare with Fig. 5 (c)). The differences between ERA5 and MERIS grid-cell broadband albedo are lower on 25 June 2007 ($\pm 0.10$), after melting has started in the whole Arctic.

Scatterplots of both grid-cell broadband albedo products for the selected days are shown in Fig. 7. To mitigate impacts on differences between both albedo products due to sea-ice edge effects, we consider only data at ERA5 sea-ice concentrations $c > 0.15$ and at distances more than 50 km from coastlines. The correlation of MERIS and ERA5 grid-cell broadband albedo varies between 0.69 and 0.79. The point clusters indicate that in most of the cases high grid-cell broadband albedo values from ERA5 are lower than respective values from MERIS observations. According to Eq. (2), the highest achievable grid-cell broadband albedo is restricted to the albedo of sea ice $\alpha_{\mathrm{seaice}}$ at $c = 1$. This restriction is clearly visible at the cluster borders at high ERA5 grid-cell broadband albedo values on 12 and 25 June. In contrast, medium grid-cell broadband albedo values from ERA5 and MERIS observations are similar, although the $\mathrm{RMSD} \approx 0.10$ is more than twice as high as the accuracy required for climate studies (0.02 - 0.05) (Henderson-Sellers and Wilson, 1983; Jacob and Olioso, 2005; Sellers et al., 1995). Low grid-cell broadband albedo values from MERIS are lower than respective values from ERA5. However, the small cluster at MERIS albedo $\alpha_{\mathrm{MERIS}} \approx 0$ can be incorrectly caused by very high melt pond fractions $s$ where the spectral albedo derivation by the MPD algorithm might fail.

## 7    Conclusions

In this study, a spectral-to-broadband conversion (STBC) has been developed empirically which can be used in combination with the Melt Pond Detector (MPD) algorithm (Zege et al., 2015) to derive broadband albedo (300 - 3000 nm) of Arctic sea ice from MERIS swath data. The empirically derived STBC has been developed with spectral and broadband albedo measured over landfast ice close to Barrow, AK, USA between April and June of the years 2008 and 2009 (Polashenski, 2011). It has been validated with airborne measured spectral and broadband albedo of drifting sea ice north of Svalbard on 25 June 2017 (Wendisch et al., 2019) (Fig. 2 (a)). Compared to the measured broadband albedo, the empirically derived STBC calculates broadband albedo with a low root mean square deviation (RMSD) of 0.02 and a high correlation of 0.98. However, low broadband albedo values are slightly underestimated. It performs more accurate than conversions proposed by Istomina et al. (2015) and Gao et al. (2004) which results in RMSDs of 0.09 and 0.05, respectively (Figs. 2 (b) and (c)).

The empirically derived STBC replaces the spectral averaging method proposed by Istomina et al. (2015) in the derivation of broadband albedo from MERIS swath data. The new broadband albedo values improve the validation of the MPD algorithm with respective airborne measurements over the southern Beaufort Sea between May and June 2008 (Birnbaum et al., 2009) (Fig. 3). The RMSD varies between 0.04 for landfast ice and 0.07 for drift ice. Both RMSDs are 0.02 lower than those when using the conversion method by Istomina et al. (2015). The higher RMSD for drift ice can result from the sea-ice drift which

displaces the airborne measurements relative to satellite observations during the time lag between both measurements. Another cause of uncertainty is the different spatial resolution of airborne and satellite measurements in combination with the more inhomogeneous surface condition of drift ice relative to landfast ice.

The new MERIS derived broadband albedo of sea ice has been compared with respective albedo values retrieved from atmospheric reanalysis data set ERA5 (ECMWF, 2019) to test the albedo parameterization used in ERA5. Both albedo products agree in the large-scale and the seasonal pattern, but differences occur spatiotemporally with RMSDs of around 0.10 (Fig. 7), exceeding the required absolute accuracy of 0.02 - 0.05 in climate models (Henderson-Sellers and Wilson, 1983; Jacob and Olioso, 2005; Sellers et al., 1995). The monthly interpolation of constant sea-ice albedo values in ERA5 based on literature values of dry snow (May, September), melting snow (June), and bare sea ice (July, August) is not suited to simulate the observed spatiotemporal variability and change in sea-ice albedo. Thus, the ERA5 broadband albedo is underestimated over multiyear ice between June and August by up to 0.1 and overestimated over first-year ice from mid-August to mid-September by up to 0.2 (Fig. 5) compared to the MERIS broadband albedo. Due to unconsidered melt ponds in ERA5, its broadband albedo over first-year ice is up to 0.1 lower than MERIS broadband albedo in June to mid-August.

As an interim solution, prognostic modeling schemes of sea-ice albedo should be developed for ERA5, which can be validated against observational products like the MERIS derived broadband albedo presented here. In order to obtain more individual values, it might be desirable to assimilate satellite derived sea-ice albedo when producing an atmospheric reanalysis. MERIS derived broadband albedo can be useful in this context, because it is not based on a priori values and is available daily even though only at daylight and under cloud free conditions.

The empirically derived STBC has been developed based on an albedo data set measured under clear-sky, cloudy, and overcast conditions. However, clouds increase the broadband albedo of snow up to 0.06 (Key et al., 2001; Grenfell and Perovich, 1984). Thus, clouds affect the accuracy of the empirically derived STBC which may contribute to the uncertainty in converting spectral-to-broadband albedo under clear-sky conditions, i. e., the only conditions when satellite data is available. The influence of this uncertainty on the satellite albedo retrieval needs further investigations. Radiative transfer simulations can be a beneficial approach in this context because they offer more flexible and specific set-ups of surface, atmosphere, and cloud types as available in short term campaigns. Another advantage of radiative transfer simulations is the calculation of different radiometric parameters for the same meteorological situation whereas measurements are done within a certain time frame in which the meteorological situation can rigorously change.

The combination of the MPD algorithm and the empirically derived STBC allows to derive broadband albedo for the complete Arctic sea ice cover under cloud free conditions between March 2002 and April 2012 from MERIS swath data. This 10 years long data set at https://www.seaice.uni-bremen.de/ can be used with the additional melt pond fraction data in Arctic climate investigations, for evaluation, validation, and data assimilation purposes, as well as in comparative studies. As a next step, swath data from OLCI (Ocean and Land Color Instrument) onboard Sentinel-3 will be implemented in the MPD algorithm to extend the broadband albedo time series of Arctic sea ice from 2016 up to date.

*Author contributions.* C. Pohl developed the spectral-to-broadband conversion, performed the computations, analysed and interpreted the results as well as outlined and wrote the manuscript with contributions from all authors. L. Istomina conceived of the original idea and provided the MERIS spectral albedo as well as codes to map the albedo data and to validate the Melt Pond Detector algorithm. S. Tietsche aided in interpreting all results related to ERA5. E. Jäkel and J. Stapf performed the measurements during the ACLOUD campaign, processed the experimental data, and contributed to the interpretation of the results. G. Spreen and G. Heygster supervised the project and aided the development of the algorithm and the comparison. All authors provided critical feedback to the manuscript.

*Competing interests.* The authors declare that no competing interests are present.

*Acknowledgements.* We gratefully acknowledge the funding by the Deutsche Forschungsgemeinschaft (DFG, German Research Foundation) – Project Number 268020496 – TRR 172, within the Transregional Collaborative Research Center "ArctiC Amplification: Climate Relevant Atmospheric and SurfaCe Processes, and Feedback Mechanisms $(AC)^3$".

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

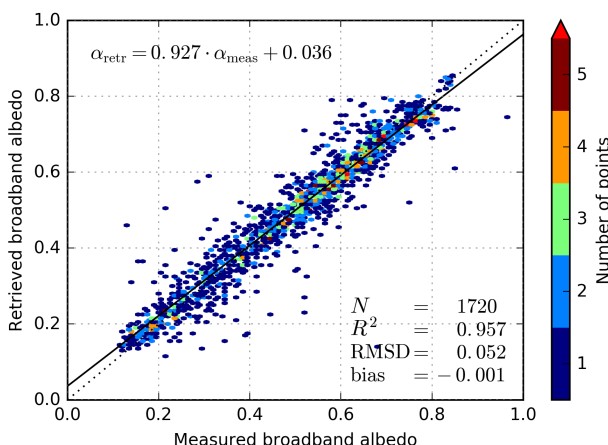

**Figure 1.** Scatterplot of measured broadband albedo and retrieved broadband albedo by applying the empirically derived STBC to measured spectral albedo. Measurements were done by Polashenski (2011) over landfast ice of different melting stages close to Barrow, AK, USA between April and June of the years 2008 and 2009. Colors indicate the number density of points. The solid and dotted lines indicate the regression and 1-to-1 line, respectively.

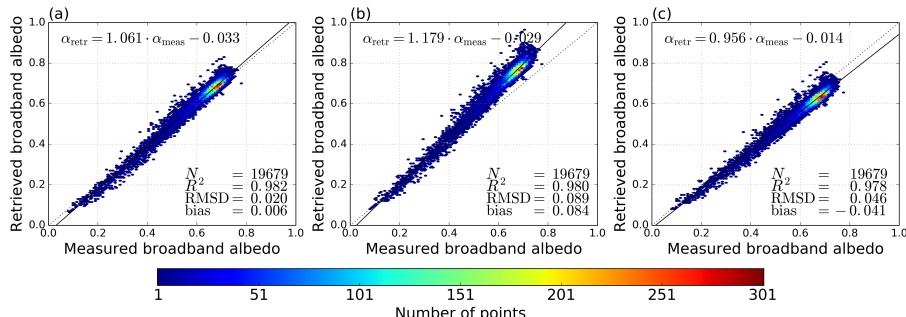

**Figure 2.** Scatterplot of measured broadband albedo and retrieved broadband albedo by applying the empirically derived STBC (a), conversion method by Istomina et al. (2015) (b), or the conversion method by Gao et al. (2004) (c) to measured spectral albedo. Measurements were done with aircraft Polar 5 over sea ice floes on 25 June 2017. Colors indicate the number density of points. The solid and dotted lines indicate the regression and 1-to-1 line, respectively.

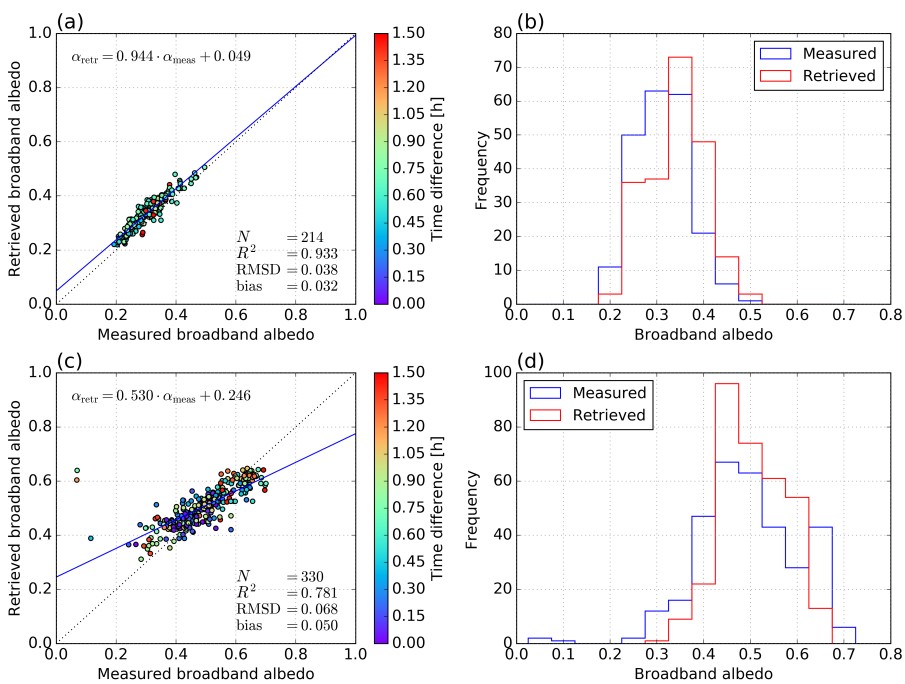

**Figure 3.** Scatterplot and histogram of airborne measured and satellite derived broadband albedo on 6 June 2008 over landfast ice (no drift) (a, b) and on 26 May and 3, 4, and 7 June 2008 over sea ice floes (possible sea-ice drift contamination) (c, d). The color of points define the time difference between airborne measurement and satellite overflight. Regression line and 1-to-1 line are shown in blue solid and black dotted, respectively.

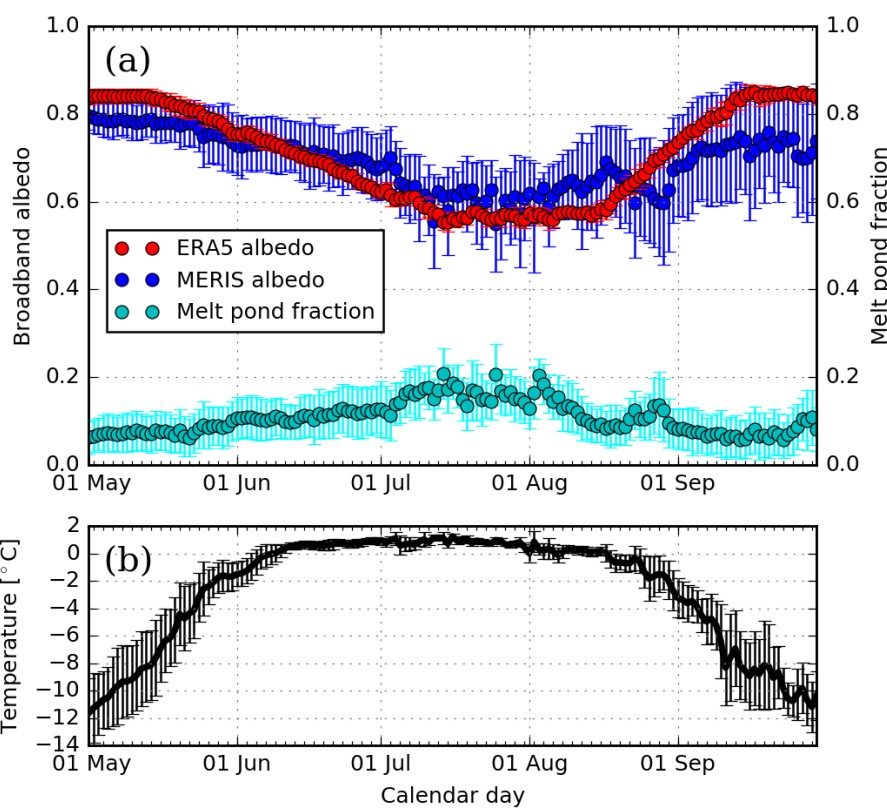

**Figure 4.** (a): Grid-cell broadband albedo from ERA5 (red) and MERIS (blue) as well as MERIS melt pond fraction (cyan) at grid-cells of 100 % sea-ice concentration according to ERA5 and at distances more than 50 km from coastlines. (b) Respective 2 m temperature from ERA5. Points and the black line show the values between 2003 and 2011 which are averaged spatially and yearly. Error bars represent their standard deviations.

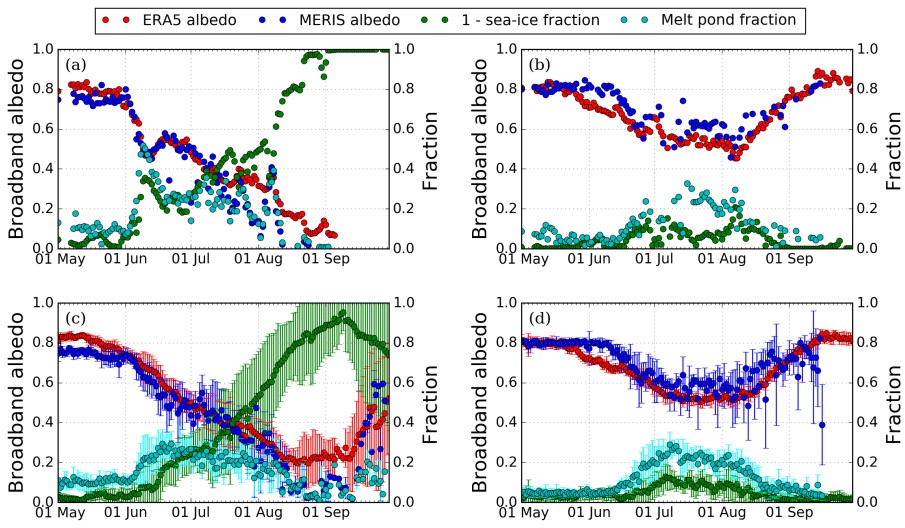

**Figure 5.** Grid-cell broadband albedo from ERA5 (red) and MERIS (blue), the complement of ERA5 sea-ice fraction (green), and the MERIS melt pond fraction (cyan) over a first-year ice area in the Beaufort Sea near Barrow (75 ° N, 155 ° W) (a, c) and a multiyear ice area north of Greenland (84.5 ° N, 35 ° W) (b, d). (a) and (b) show the values between May and September 2007. (c) and (d) show the values that were averaged over the years from 2003 to 2011 (points) and their standard deviations (error bars).

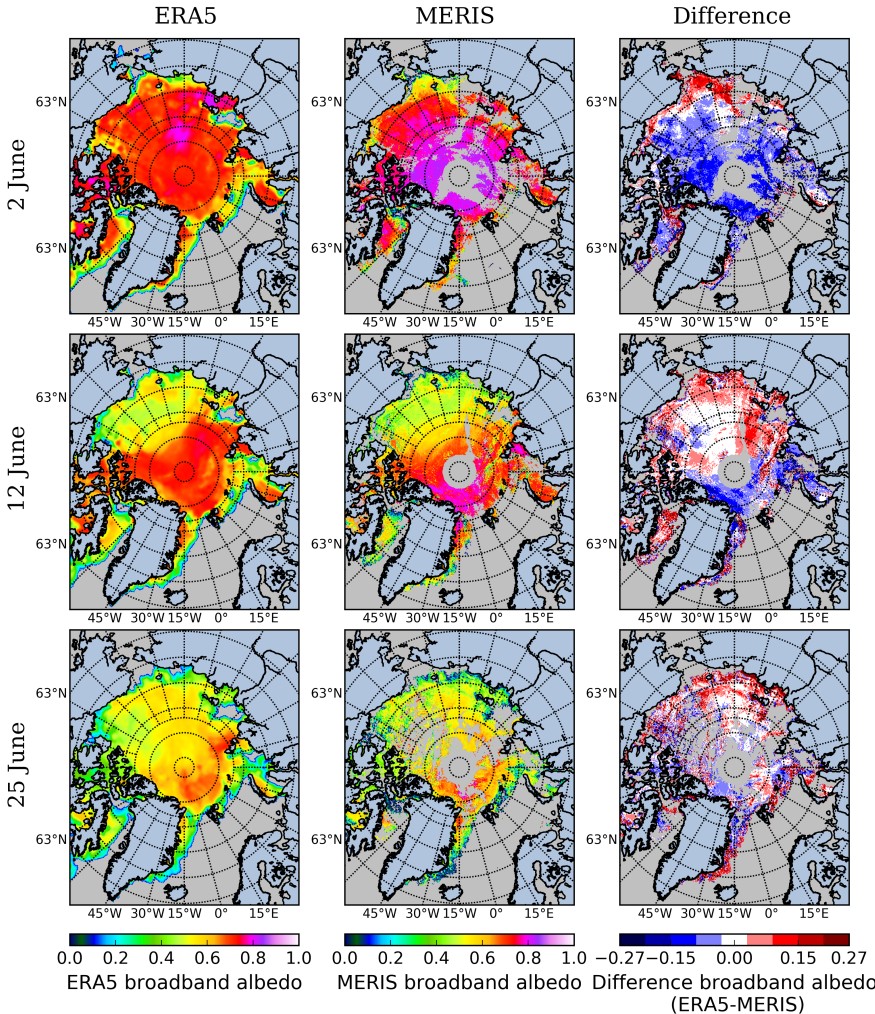

**Figure 6.** Daily averaged grid-cell broadband albedo of Arctic sea ice derived from ERA5 irradiances (left) and from MERIS observations (middle) and the difference between both albedo products (right). The data are from 2 (top, before melt onset), 12 (middle, at melt onset), and 25 (bottom, after melt onset) June 2007. Cloud contaminated areas and the satellite pole hole in the MERIS data set are shown in gray. Data are gridded using the NSIDC (National Snow and Ice Data Center) polar stereographic projection.

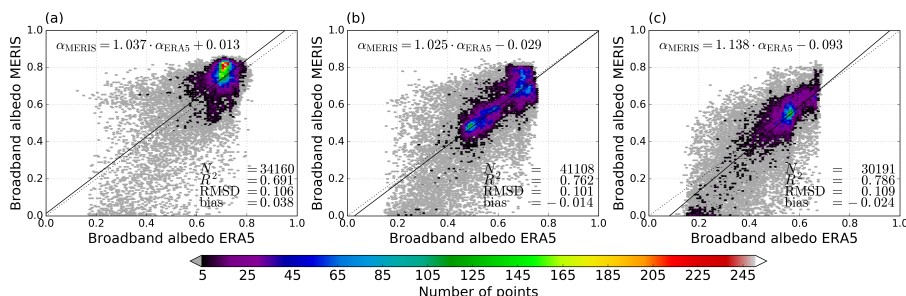

**Figure 7.** Comparison of ERA5 and MERIS derived grid-cell broadband albedo of Arctic sea ice on selected days of Fig. 6 ((a): 2 June 2007, before melt onset; (b): 12 June 2007, at melt onset; (c): 25 June 2007, after melt onset). Regression line and 1-to-1 line are shown in black solid and dotted, respectively.

**Table 1.** Major surface broadband albedo products derived from satellite observations.

| Data set name | Instrument | Platform | Spatial resolution | Temporal resolution | Temporal coverage | Waveband [$\mu m$] | Retrieval method | References |
|---|---|---|---|---|---|---|---|---|
| - | MODIS | Terra, Aqua | 1 km / 0.05 ° | 8 days | 2000 – now | 0.3 - 5 | kernel driven BRDF, NTBC | Lucht et al. (2000), Schaaf et al. (2002) |
| APP-x | AVHRR | NOAA POES | 5 km | Twice daily (Arctic region) | 1982 – now | 0.28 - 4 | NTBC, ARC | Key et al. (2016) |
| - | MERIS | ENVISAT | 1 km / 0.05 ° | 1 day | 2002–2012 | 0.4 - 0.9 | Analytic solution of RTT, STBC | Zege et al. (2015), Is-tomina et al. (2015) |
| - | VIIRS | S-NPP | 1 km | 1 day / 16 days | 2011 – now | 0.4 - 4 | Direct estimation | Wang et al. (2013), Zhou et al. (2016) |
| - | OLI | Landsat | 30 m | 16 days | 2013 – now | 0.3 - 3 | Direct estimation | He et al. (2018a, b) |
| - | TM/ETM+ | Landsat | 30 m | 16 days | 1982 – now | 0.3 - 3 | Direct estimation | He et al. (2018a, b) |
| CLARA-A2 SAL | AVHRR | NOAA, Metop | 0.25 ° | 5 days / 1 month | 1982-2015 | 0.25 - 2.5 | NTBC, ARC | Riihelä et al. (2013), Karlsson et al. (2017) |
| Glob-Albedo | VEGETATION, AATSR, MERIS, MODIS | SPOT, EN-VISAT, Terra, Aqua | 1 km / 0.05 °, 0.5 ° | 16 days / 1 month | 1998-2011 | 0.3 - 3 | NTBC, kernel driven BRDF | Muller et al. (2012), Lewis et al. (2013) |
| GLASS | MODIS, AVHRR | Terra, Aqua, NOAA | 1 km / 0.05 ° | 1 day / 8 days | 1981-2012 | 0.3 - 3 | Direct estimation | Liu et al. (2013), Liang et al. (2013) |
| CERES-EBAF | CERES | Terra, Aqua, S-NPP, NOAA | 1 ° | 1 day / 1 month | 2000 - now | 0.3 - 5 | RTT | CERES-EBAF (2014), Loeb et al. (2018) |
| ISCCP | VISSR, MIR, AVHRR | GMS, GOES, INSAT, ME-TEOSAT, NOAA | 2.5 ° | 1 month | 1983-2009 | 0.2 - 5 | RTT | Zhang et al. (1995, 2004), Schiffer and Rossow (1983) |

**Table 2.** Coefficients $k_i$ in Eq. (1) to calculate MERIS broadband albedo from spectral albedo at the six wavelengths $400, 500, 600, 700, 800,$ and 900 nm (for Gao et al. (2004) from the narrowband albedo at channels with the center wavelengths $490, 560, 665,$ and 865 nm).

|  | $k_0$ | $k_1$ | $k_2$ | $k_3$ | $k_4$ | $k_5$ | $k_6$ |
|---|---|---|---|---|---|---|---|
| Empirically derived | 0.0000 | 0.9337 | -2.0856 | 2.9125 | -1.6231 | 0.6750 | 0.0892 |
| Istomina et al. (2015) | 0.0000 | 0.1666 | 0.1666 | 0.1666 | 0.1666 | 0.1666 | 0.1666 |
| Gao et al. (2004) | 0.0149 | 0.1587 | -0.2463 | 0.5442 | 0.3748 | 0.0000 | 0.0000 |

**Table 3.** Times of airborne measurements (Polar 5) and satellite overflights (MERIS on ENVISAT) in UTC for selected days during the MELTEX campaign in 2008.

|         | 26 May 2008 | 3 June 2008 | 4 June 2008 | 6 June 2008 | 7 June 2008 |
|---------|-------------|-------------|-------------|-------------|-------------|
| Polar 5 | 20:45-21:48 | 17:00-19:46 | 19:14-23:24 | 19:01-21:55 | 17:08-20:17 |
| MERIS   | 20:46       | 19:54       | 21:02       | 20:00       | 21:08       |