# Peer review of "Broadband albedo of Arctic sea ice from MERIS optical data"

_The Cryosphere, 2019_

## Referee Comment (RC1) · Anonymous Referee #1 · 10 Jun 2019

The characterization of the radiative properties of Arctic sea ice is of substantial importance for many aspects of climate modeling and monitoring. Thus the present effort to improve upon the MERIS record is welcomed, although the record's relatively short length does place some restriction on its use. Also, the manuscript's comparison against ERA5 sea ice albedo is welcomed, although the rather limited scope of the comparison leaves the reader with rather many open questions. This reviewer therefore suggests adding more meat into that section, as the other part of the manuscript (updated NBC) is somewhat light in content for a full Cryosphere paper in itself.

Major comments

a. Please elaborate on the ERA5 comparison. What are the likely causes for the discrepancies? What is the associated uncertainty in the MERIS estimates, how much of

the difference is explainable through them? Are there regionally or temporally changing drivers behind the differences?

b. Both training and evaluation of the method appear to be based on early-season ice, May-June. Some discussion is warranted on whether or not this implies issues in the determination of late-season ice cover, given the surface changes incurred by e.g. melt pond draining or surface refreezing.

c. I approve of the airborne measurement comparison, but have you evaluated against full-summer albedo observations, such as those available from the Tara expedition of 2007-2008? If not, why not as that is in your study period?

d. While I understand the brevity in method description given the authors' past works on the topic, a short summary of a couple of sentences describing the principles behind e.g. the BRDF model for the highly heterogeneous sea ice cover and the choice of melt pond optical properties for the varied types of melt ponds seems in order in section 2.1 to facilitate context for the readers.

e. Is the updated MERIS sea ice albedo dataset available somewhere? The manuscript implies it, but no access method is given anywhere.

Minor comments

2, 9-26: For the legacy orbiter datasets, APP-x and GLASS are mentioned but not CM SAF CLARA. Why the omission?

9, 19-22: For clarity, please mention what the Ebert-Curry parameterization is based on (air temperature, or?)

11, 22: by -> up to, depends on the amount of diffuse radiation in the downwelling flux and the surface BRDF.

---

## Referee Comment (RC2) · Anonymous Referee #2 · 11 Jul 2019

Mapping broadband albedo of Arctic sea ice from remote sensing data is critical important for cryosphere and global climate change studies. However, the reflectance anisotropic effects of Arctic sea ice and melt ponds have not been well considered in current available broadband albedo products. Thus, it is still a challenging topic for developing long-term and high quality Arctic sea ice broadband surface albedo dataset. In this paper, the authors provided a new spectral-to-broadband conversion (STBC) method which can obtain much accurate broadband albedo compared with the conversion coefficients provided by other literatures. In additions, the authors also compared the Artic sea ice albedo derived from ERA5 and MERIS data. From my point of view, the parts for describing the experiments and methods are solid and credible. However, the discussions for the STBC method and comparison with reanalysis data are too

short. I would like to suggest the authors to add more discussions and comparisons before developing it into a final publish paper. Major comments: 1. The STBC method has not considered the variations of atmospheric conditions and solar/view zenith angle. Please add a discussion about the uncertainties of the STBC method proposed by this paper. References: Liang S. (2001). Narrowband to broadband conversions of land surface albedo I: Algorithms. Remote Sensing of Environment, 76(2), 213-238 Liang S., Shuey C. J., Russ A. L., et al.(2003). Narrowband to broadband conversions of land surface albedo: II. Validation. Remote Sensing of Environment, 84(1), 25-41 2. The STBC method proposed by this paper has not considered the difference between the black-sky and white-sky albedo. Please provide different conversion coefficients or add a discussion about it. 3. I suggest to add a short describing for the Melt Pond Detector (MPD) algorithm and the procedure for generating the spectral albedo and melt pond fraction of sea ice. 4. It is necessary to validate the broadband albedo of Arctic sea ice from MERIS data with in situ albedo measurements, e.g., TARA data. 5. It is necessary to compare the broadband albedo of Arctic sea ice from MERIS data with other broadband albedo products, such as CLARA, VIIRS, and GLASS. 6. Why there are differences between albedo derived from ERA5 and MERIS data? Is there any relationship between differences and melt ponds fraction? Please add a discussion about it. Minor comments: 1. Page 10, Line 11. Why the $k_0$ is set to 0? 2. Page 9, Line 25. The variable c has not been described. 3. Page 10, Line 31. The word "then" should be "than".

---

## Author Comment (AC1) · 2 Sep 2019

Dear Referee 1,

We thank for the time and efforts you spent reading our manuscript.

Below you can find your comments, our response and our changes in the manuscript point by point.

Best wishes, Christine Pohl

**The characterization of the radiative properties of Arctic sea ice is of substantial importance for many aspects of climate modeling and monitoring. Thus the present effort to improve upon the MERIS record is welcomed, although the record's relatively**

short length does place some restriction on its use. Also, the manuscript's comparison against ERA5 sea ice albedo is welcomed, although the rather limited scope of the comparison leaves the reader with rather many open questions. This reviewer therefore suggests adding more meat into that section, as the other part of the manuscript (updated NBC) is somewhat light in content for a full Cryosphere paper in itself.

Major comments: a. Please elaborate on the ERA5 comparison. What are the likely causes for the discrepancies? What is the associated uncertainty in the MERIS estimates, how much of the difference is explainable through them? Are there regionally or temporally changing drivers behind the differences?

We expand the comparison of the ERA5 and MERIS derived broadband albedo by introducing temporal differences between both products at ERA5 sea-ice concentration of 100 % and at two different areas (on first-year ice in the Beaufort Sea near Barrow, 75° N, 155° W, and on multiyear ice north of Greenland, 84.5° N, 35° W) in the summer month (May – September) 2007. We highlight systematical offsets for both mentioned areas by averaging these differences between the years 2003 and 2011. By introducing the melt pond fraction from MERIS data and sea-ice concentration from ERA5 data, we can identify several causes for the differences, e.g. undetected leads (open water) in MPD algorithm of MERIS, unconsidered melt ponds in ERA5, and the albedo parameterization in ERA5. The quantitative estimation of MERIS broadband albedo uncertainties is difficult due to their complexity. Instead, we introduce a new section about uncertainties generating with the spectral-to-broadband conversion between the sections "Results" and "Improved melt pond detection validation".

-> We completely revised the Section 6: Comparison between broadband albedo from satellite and atmospheric reanalysis. -> We newly introduced the Section 4: Uncertainties due to differing variables along the spectral-to-broadband conversion.

**b. Both training and evaluation of the method appear to be based on early-season ice, May-June. Some discussion is warranted on whether or not this implies**

issues in the determination of late-season ice cover, given the surface changes incurred by e.g. melt pond draining or surface refreezing.

Discussion in Section 4 (p. 7 l. 27-30):

The STBC is based on various sea-ice surface types in the early season (April - June) mentioned at the end of Sect. 2.2. Nevertheless, it can be applied to later seasonal (July-September) Arctic surface type as well, as those surface types like snow covered sea-ice, white ice, and refrozen surface types are similar to the surface types included in the training data-set (snow layers and frozen melt ponds). Therefore, the STBC can be used for all Arctic sea-ice surface types.

**c. I approve of the airborne measurement comparison, but have you evaluated against full-summer albedo observations, such as those available from the Tara expedition of 2007-2008? If not, why not as that is in your study period?**

It is difficult to compare the Tara data to MERIS broadband albedo because of the different spatial resolution. While the resolution of MERIS data is $(12.5 \text{ km})^2$, the TARA data were measured at a local place. Discrepancies between both albedo products will occur due to uncertainties in the representativeness of each in-situ TARA measurement for the corresponding MERIS pixel. In contrast, airborne measurements can average the local surface properties for each MERIS pixel, making the comparison from airborne to MERIS derived broadband albedo more significant. Therefore, we omit the comparison of MERIS broadband albedo to TARA broadband albedo.

**d. While I understand the brevity in method description given the authors' past works on the topic, a short summary of a couple of sentences describing the principles behind e.g. the BRDF model for the highly heterogeneous sea ice cover and the choice of melt pond optical properties for the varied types of melt ponds seems in order in section 2.1 to facilitate context for the readers.**

We give more details about the MERIS MPD algorithm in Sect. 2.1. (p. 4 l. 14-28):

For each classified sea-ice grid-cell, ice and pond parameters (optical thickness of pond and ice, scattering coefficient of ice, effective ice grain size, and absorption coefficient of yellow pigments) as well as a pond fraction are initialized. From those, the white ice and melt pond BRDF are calculated based on the asymptotic solution for optically thick layers (Zege et al., 1991). In case of the melt pond BRDF calculation, the optically thick layer is referred to the melt pond bottom. The reflection and transmission at air-water interface is determined by Fresnel's law. The surface BRDF is calculated as a linear combination of both BRDF values weighted by the pond fraction. From surface BRDF and the atmospheric reflectance and transmittance calculated by the radiative transfer code RAY (Tynes et al., 2001), the radiances at MERIS channels at top of atmosphere are derived based on the atmospheric correction method by Tanre et al. (1983). In an iterative process based on the Newton-Raphson method (Press and Flannery, 1993), the difference of measured and calculated MERIS radiances is minimized as a function of the ice and pond parameters, and of the melt pond fraction. From resulting ice and pond parameters, the spectral black-sky albedo (directional-hemispherical albedo (Schaepman-Strub et al., 2006)) at wavelengths $\lambda_i$ = 400; 500; 600; 700; 800; and 900 nm are calculated. The output of the MPD algorithm are the spectral albedo at mentioned six wavelengths, the melt pond fraction and the estimated retrieval error for each MERIS swath data grid-cell. Daily averages are created by gridding and averaging the output of each MERIS swath on a 12.5 km NSIDC (National Snow and Ice Data Center) polar stereographic projection.

**e. Is the updated MERIS sea ice albedo dataset available somewhere? The manuscript implies it, but no access method is given anywhere.**

The updated MERIS data set will be published at https://seaice.uni-bremen.de/melt-ponds/. A respective comment will be given in the manuscript.

**Minor comments 2, 9-26: For the legacy orbiter datasets, APP-x and GLASS are mentioned but not CMSAF CLARA. Why the omission?**

We introduced CLARA-A2 SAF in Table 1 and corresponding text:

p. 2, l. 23-26:

A 34 year time series of black-sky surface albedo (SAL) as part of the second edition of the cloud, albedo, and surface radiation data set (CLARA-A2) has been derived from AVHRR measurements onboard NOAA and Metop (Meteorological Operational) satellites (Riihelä et al., 2013; Karlsson et al., 2017).

p. 31 (table):

Data set name: CLARA-A2 SAL Instrument: AVHRR Platform: NOAA, Metop Spatial resolution: 0.25° Temporal resolution: 5 days / 1 month Temporal coverage: 1982 – 2015 Waveband: 0.25 – 2.5 Retrival method: NTBC, ARC References: Riihelä et al. (2013), Karlsson et al. (2017)

**9, 19-22: For clarity, please mention what the Ebert-Curry parameterization is based on (air temperature, or?)**

The Ebert-Curry parameterization is based on the solar zenith angle and the spectral dependency of reflection. The latter is only considered in the ERA5 parameterization:

p.11 l. 3-4: The sea-ice albedo is simply prescribed as constants following the albedo values given in Ebert and Curry (1993) and considers the spectral variation.

**11, 22: by -> up to, depends on the amount of diffuse radiation in the down-welling flux and the surface BRDF.**

The mentioned sentence is improved (p. 16, l. 12):

However, clouds increase the broadband albedo of snow up to 0.06 (Key et al., 2001; . . .)

Please also note the supplement to this comment:
https://www.the-cryosphere-discuss.net/tc-2019-62/tc-2019-62-AC1-supplement.pdf

---

## Author Comment (AC2) · 2 Sep 2019

Dear Referee 2,

We thank for the time and efforts you spent reading our manuscript.

Below you can find your comments, our response and our changes in the manuscript point by point.

Best wishes, Christine Pohl

**Mapping broadband albedo of Arctic sea ice from remote sensing data is critical important for cryosphere and global climate change studies. However, the reflectance anisotropic effects of Arctic sea ice and melt ponds have not been well considered incurrent available broadband albedo products. Thus, it is still a challenging topic for developing long-term and high quality Arctic sea ice broadband surface albedo dataset. In this paper, the authors provided a new spectral-to-broadband conversion (STBC) method which can obtain much accurate broadband albedo compared with the conversion coefficients provided by other literatures. In additions, the authors also compared the Artic sea ice albedo derived from ERA5 and MERIS data. From my point of view, the parts for describing the experiments and methods are solid and credible. However, the discussions for the STBC method and comparison with reanalysis data are too short. I would like to suggest the authors to add more discussions and comparisons before developing it into a final publish paper.**

Major comments: 1. The STBC method has not considered the variations of atmospheric conditions and solar/view zenith angle. Please add a discussion about the uncertainties of the STBC method proposed by this paper. References: Liang S. (2001). Narrowband to broadband conversions of land surface albedo I: Algorithms. Remote Sensing of Environment, 76(2), 213-238Liang S., Shuey C. J., Russ A. L., et al.(2003). Narrowband to broadband conversionsof land surface albedo: II. Validation. Remote Sensing of Environment, 84(1), 25-41

A discussion about the variability of atmospheric condition and solar zenith angle is added in the newly introduced section "4 Uncertainties due to differing variables along the spectral-to-broadband conversion". We refer to the reference Liang (2001) but omit the reference Liang et al. (2003), because they don't consider surface types similar to sea ice and neither solar zenith angle nor atmospheric variation is investigated.

p. 8 l. 14-21: According to Liang (2001), a wide range of atmospheric conditions and solar zenith angles are required to develop the STBC generally applicable to any spectral albedo measurement in the Arctic. We can confirm a high variability of the former via available aerosol optical depth from AERONET (Aerosol Robotic Network) data measured in Barrow, AK, USA, for the days of albedo measurements by Polashenski (2011) when the sun disk was visible. At those days, the solar zenith angle can vary by

35 °. However, due to the lack of the exact times of albedo measurements, we cannot determine the actual variability in the solar zenith angle. Since variations in the atmospheric visibility and the solar zenith angle influence the broadband albedo (Liang, 2001) only marginally, they play a minor role in the uncertainties of the empirically derived STBC. Additionally, those effects are overlaid by the cloud effects mentioned above.

**2. The STBC method proposed by this paper has not considered the difference between the black-sky and white-sky albedo. Please provide different conversion coefficients or add a discussion about it.**

We introduced a discussion about discrepancies between black- and white sky (blue-sky) albedo in the newly introduced section "4 Uncertainties due to differing variables along the spectral-to-broadband conversion".

p. 8 l. 27-39:

Although the STBC has been derived for blue-sky albedo (bihemispherical reflectance (Schaepman-Strub et al., 2006)) in Sect. 2.2, it shall be applied to the MERIS derived spectral albedo which is referred to as black-sky albedo (directional hemispherical reflectance). Manninen et al. (2012) estimated the discrepancy between black- and blue-sky broadband albedo for 87 individual surface reflectance spectra from USGS (U. S. Geological Survey) Spectroscopy Lab data which includes, i. a., snow, ice and water spectra. It is typically lower than 0.05 at solar zenith angles around 60 ° and at aerosol optical depths up to 1 at 440 nm. We expect that the aerosol optical depth is considerably lower in the Arctic such that the difference between black- and blue-sky albedo decreases and seems to be negligible against the cloud effect. Hence, we decide not to adjust any albedo measurement for the blue-sky / black-sky discrepancy.

**3. I suggest to add a short describing for the Melt Pond Detector (MPD) algorithm and the procedure for generating the spectral albedo and melt pond fraction of sea ice.**

We give more details about the MERIS MPD algorithm in Sect. 2.1. (p. 4 l. 14-28):

For each classified sea-ice grid-cell, ice and pond parameters (optical thickness of pond and ice, scattering coefficient of ice, effective ice grain size, and absorption coefficient of yellow pigments) as well as a pond fraction are initialized. From those, the white ice and melt pond BRDF are calculated based on the asymptotic solution for optically thick layers (Zege et al., 1991). In case of the melt pond BRDF calculation, the optically thick layer is referred to the melt pond bottom. The reflection and transmission at air-water interface is determined by Fresnel's law. The surface BRDF is calculated as a linear combination of both BRDF values weighted by the pond fraction. From surface BRDF and the atmospheric reflectance and transmittance calculated by the radiative transfer code RAY (Tynes et al., 2001), the radiances at MERIS channels at top of atmosphere are derived based on the atmospheric correction method by Tanre et al. (1983). In an iterative process based on the Newton-Raphson method (Press and Flannery, 1993), the difference of measured and calculated MERIS radiances is minimized as a function of the ice and pond parameters, and of the melt pond fraction. From resulting ice and pond parameters, the spectral black-sky albedo (directional-hemispherical albedo (Schaepman-Strub et al., 2006)) at wavelengths $\lambda_i$ = 400; 500; 600; 700; 800; and 900 nm are calculated. The output of the MPD algorithm are the spectral albedo at mentioned six wavelengths, the melt pond fraction and the estimated retrieval error for each MERIS swath data grid-cell. Daily averages are created by gridding and averaging the output of each MERIS swath on a 12.5 km NSIDC (National Snow and Ice Data Center) polar stereographic projection.

**4. It is necessary to validate the broadband albedo of Arctic sea ice from MERIS data with in situ albedo measurements, e.g., TARA data.**

It is difficult to compare the Tara data to MERIS broadband albedo because of the different spatial resolution. While the resolution of MERIS data is (12.5 km)$^2$, the TARA data were measured at a local place. Discrepancies between both albedo products will occur due to uncertainties in the representativeness of each in-situ TARA measurement for the corresponding MERIS pixel. In contrast, airborne measurements can average the local surface properties for each MERIS pixel, making the comparison from airborne to MERIS derived broadband albedo more significant. Therefore, we omit the comparison of MERIS broadband albedo to TARA broadband albedo.

**5. It is necessary to compare the broadband albedo of Arctic sea ice from MERIS data with other broadband albedo products, such as CLARA, VIIRS, and GLASS.**

Comparison to other satellite products is a challenge because of different properties of the derived satellite albedo product: Available VIIRS albedo starts from 2011 and is not convenient for a "long-term" albedo comparison since MERIS broadband albedo is only available between 2003 and 2011. GLASS provides broadband albedo only at solar noon and is not a daily averaged albedo product as MERIS albedo. CLARA-SAL albedo has the best potential to be compared to MERIS albedo, although it is available in a 5 day resolution. Unfortunately, the CLARA-SAL albedo over sea ice contains uncertainties based on its own retrieval method such as it cannot be used as a reference albedo product in the comparison of MERIS and ERA5 broadband albedo. A discussion about the uncertainties of CLARA-SAL exceeds the scope of our manuscript. Therefore, we omit the comparison with CLARA-SAL albedo data.

**6. Why there are differences between albedo derived from ERA5 and MERIS data? Is there any relationship between differences and melt ponds fraction? Please add a discussion about it.**

We expand the comparison of the ERA5 and MERIS derived broadband albedo by introducing temporal differences between both products at ERA5 sea-ice concentration of 100 % and at two different areas (on first-year ice in the Beaufort Sea near Barrow, 75° N, 155° W, and on multiyear ice north of Greenland, 84.5° N, 35° W) in the summer month (May – September) 2007. We highlight systematical offsets for both mentioned areas by averaging these differences between the years 2003 and 2011.

By introducing the melt pond fraction from MERIS data and sea-ice concentration from ERA5 data, we can identify several causes for the differences, e.g. undetected leads (open water) in MPD algorithm of MERIS, unconsidered melt ponds in ERA5, and the albedo parameterization in ERA5.

-> We completely revised the Section 6: Comparison between broadband albedo from satellite and atmospheric reanalysis.

**Minor comments: 1. Page 10, Line 11. Why the k0 is set to 0?**

Explanation is introduced at p. 5 l. 25-26:

The coefficient k0 was preset to 0 in order to warrant the condition alpha_bb = alpha_(lambda_i) = 0.

**2. Page 9, Line 25. The variable c has not been described.**

The name of the variable is introduced, now (p. 11 l. 10):

. . ., with c representing the sea-ice concentration from ERA5 generated by Copernicus Climate Change Service (2017).

**3. Page 10, Line 31. The word "then" should be "than".**

It is corrected (p. 15 l. 19):

It performs more accurate than conversions proposed by . . .

Please also note the supplement to this comment:
https://www.the-cryosphere-discuss.net/tc-2019-62/tc-2019-62-AC2-supplement.pdf

[Figure]

**Supplement:**

[revised manuscript text omitted]

---

## Author Response (AR2)

Dear reviewers, dear editor,

thank for the time and efforts for reading our manuscript. There was one remaining question of reviewer 2, which we will answer below. Additionally, we are providing a list of relevant changes made in the manuscript.

Best wishes,
Christine Pohl and Co-authors

Question of reviewer 2:

As the fraction of melt pond was not considered by the ERA5 reanalysis data, the albedo of arctic sea-ice region should be overestimated. However, the results of this paper showed that arctic albedo in summer was underestimated by the ERA5 compared with MERIS data. Why? Please explain it in revised manuscript.

p.13 line 1-4:
Despite existing melt ponds in a grid-cell from end of June to end of August, the MERIS derived albedo $\alpha_{\mathrm{MERIS}}$ can be higher than the broadband albedo from ERA5 $\alpha_{\mathrm{ERA5}}$. However, the latter remains within the standard deviation of $\alpha_{\mathrm{MERIS}}$, implying that the literature based albedo value of bare ice $\alpha_{\mathrm{seaice}}$ is reasonable but slightly low-biased for surface types consisting of a mixture of melt ponds, wet snow, and bare ice.

Additional changes in the manuscript:

– Provision of a link to the new broadband albedo data set (as mentioned in the last review round)

– Introducing an additional reference concerning the MPD algorithm.

[revised manuscript text omitted]